# Maternal prenatal anxiety and depression and trajectories of cardiometabolic risk factors across childhood and adolescence: a prospective cohort study

Karen Matvienko-Sikar [1], Kate O' Neill [1], Abigail Fraser,[2] Catherine Hayes [3], Laura Howe,[2] Anja C Huizink,[4] Patricia M Kearney [1], Ali Khashan,[1,5] Sarah A Redsell,[6] Linda M O'Keeffe[1,2,7]

For numbered affiliations see end of article.

**Correspondence to**
Dr Karen Matvienko-Sikar;
karen.msikar@ucc.ie

## ABSTRACT

**Objectives** Quantifying long-term offspring cardiometabolic health risks associated with maternal prenatal anxiety and depression can guide cardiometabolic risk prevention. This study examines associations between maternal prenatal anxiety and depression, and offspring cardiometabolic risk from birth to 18 years.

**Design** This study uses data from the Avon Longitudinal Study of Parents and Children (ALSPAC) cohort.

**Participants** Participants were 526–8606 mother–offspring pairs from the ALSPAC cohort.

**Setting** British birth cohort set, Bristol, UK.

**Primary and secondary outcomes** Exposures were anxiety (Crown-Crisp Inventory score) and depression (Edinburgh Postnatal Depression Scale score) measured at 18 and 32 weeks gestation. Outcomes were trajectories of offspring body mass index; fat mass; lean mass; pulse rate; glucose, diastolic and systolic blood pressure (SBP); triglycerides, high-density lipoprotein cholesterol and non-high-density lipoprotein cholesterol and insulin from birth/early childhood to 18 years. Exposures were analysed categorically using clinically relevant, cut-offs and continuously to examine associations across the distribution of prenatal anxiety and depression.

**Results** We found no strong evidence of associations between maternal anxiety and depression and offspring trajectories of cardiometabolic risk factors. Depression at 18 weeks was associated with higher SBP at age 18 (1.62 mm Hg (95% CI 0.17 to 3.07). Anxiety at 18 weeks was also associated with higher diastolic blood pressure at 7 years in unadjusted analyses (0.70 mm Hg (95% CI 0.02 to 1.38)); this difference persisted at age 18 years (difference at 18 years; 0.89 mm Hg (95% CI 0.05 to 1.73). No associations were observed for body mass index; fat mass; lean mass; pulse rate; glucose; triglycerides, high-density lipoprotein cholesterol and non-high-density lipoprotein cholesterol and insulin.

**Conclusions** This is the first examination of maternal prenatal anxiety and depression and trajectories of offspring cardiometabolic risk. Our findings suggest that prevention of maternal prenatal anxiety and depression

### Strengths and limitations of this study

► This study presents a prospective measurement of anxiety and depression two times during pregnancy, including analysis of repeated measures of 11 key cardiometabolic risk factors from birth to 18 years.

► Use of multilevel models accounting for clustering of repeated measures within individuals and correlation between measures over time.

► Included participants tend to be predominantly white women and were more advantaged than those excluded due to missing exposure, confounder or outcome data, limiting generalisability.

► Anxiety and depression exposures were self-reported.

may have limited impact on offspring cardiometabolic health across the first two decades of life.

## INTRODUCTION

Maternal prenatal anxiety and depression are estimated to affect up to 25% of women.[1] Maternal prenatal anxiety and depression can result from a range of general and pregnancy-specific factors.[2] Maternal prenatal anxiety and depression are associated with adverse offspring outcomes such as low birth weight, short gestational age,[3] obstetric complications[4] and poor offspring developmental and health outcomes.[5 6] Adverse offspring outcomes may result from intrauterine programming involving endocrine, inflammatory and immunological processes.[6–8] Epigenetic changes[8] and dysregulation of the maternal hypothalamic pituitary adrenal (HPA) axis are suggested to result from prenatal distress[9]; dysregulated prenatal HPA axis activity may programme foetal HPA axis activity, reactivity and later health outcomes.[6 9]

It is recognised that there may be 'critical windows' of vulnerability during which exposure to prenatal anxiety and depression are particularly deleterious to offspring health.[10 11] However, chronic anxiety and/or depression (experienced over a prolonged period) during pregnancy may also lead to adverse offspring outcomes.[7] It is further argued that such pathways, as well as behavioural and environmental mechanisms, increase risk of unfavourable offspring cardiometabolic health outcomes, including increased adiposity and poor cardiovascular function.[6 12–14]

To date, inconsistent associations have been observed between maternal prenatal anxiety and depression, and child and adolescent cardiometabolic risk factors, including high blood pressure,[12 15] increased insulin resistance[12 14] and overweight.[13 16–19] Little is known about the effects of timing of prenatal maternal anxiety and depression exposures on offspring cardiometabolic outcomes, though there is some evidence that later exposures may confer increased risk.[13] There is some limited evidence for differential associations by 'type' of exposure, such as type of psychological distress exposure (e.g., stress, anxiety and depression), particularly for anthropometric outcomes,[14] though this is not well understood. Studies examining associations of prenatal anxiety and depression with cardiometabolic health outcomes have predominantly examined associations with outcomes at a single time point.[12 14] Examining early life trajectories of cardiometabolic health outcomes provides insights into if and when associations emerge during childhood and adolescence, and whether associations persist over time. This is particularly important as cardiovascular risk originates early in life and can track to adulthood.[20] Such examinations are needed to highlight potential mechanisms and inform the nature and timing of prevention efforts for both maternal prenatal anxiety and depression, and offspring cardiometabolic risk.

The objective of this study is to examine associations of maternal anxiety and depression with offspring cardiometabolic health outcome trajectories from birth to 18 years using data from the Avon Longitudinal Study of Parents and Children (ALSPAC).

## METHODS
### Study participants
ALSPAC is a prospective birth cohort study in Southwest England.[20 21] Pregnant women resident in one of the three Bristol-based health districts with an expected delivery date between 1 April 1991 and 31 December 1992 were invited to participate. The study has been described elsewhere in detail.[20 21] ALSPAC initially enrolled a cohort of 14 451 pregnancies, from which 13 761 women provided informed consent and had 13 867 live births. Research clinics were held when the offspring were approximately 7, 9, 10, 11, 13, 15 and 18 years old. The study website contains details of all the data that are available through a fully searchable data dictionary http://wwwbristolacuk/alspac/researchers/our-data/.

### Study exposures
We derived and separately analysed associations between different indices of maternal prenatal anxiety and depression and trajectories of cardiometabolic health outcomes.

#### Maternal self-reported prenatal anxiety
Maternal prenatal anxiety was measured at 18 and 32 weeks gestation using the eight items from the anxiety subscale of Crown Crisp Experiential Index[22]; further details in online supplemental methods S1. For our primary analysis, a score ≥85th percentile (≥85th percentile=8) was used to define anxiety.[6] We then categorised prenatal anxiety as 'anxiety at 18 weeks', 'anxiety at 32 weeks', 'anxiety at both time points' and 'anxiety at neither time point' (reference group). In secondary analyses, anxiety was analysed as a continuous measure at 18 and 32 weeks separately and taking the mean of anxiety scores at 18 and 32 weeks to explore associations with offspring cardiometabolic health across the entire distribution (to examine associations at preclinical and clinical levels).

#### Maternal self-reported depression
Maternal prenatal depression was measured at 18 and 32 weeks gestation using the Edinburgh Postnatal Depression Scale (EPDS[23]); further details in online supplemental methods S1. For our primary analysis, a score of ≥13 was used to define clinical depression.[20] We categorised maternal prenatal depression as 'depression at 18 weeks', 'depression at 32 weeks', 'depression at both time points' and 'depression at neither time point' (reference group). The continuous measure of depression using the EPDS was also examined. In secondary analyses, depression was analysed as a continuous measure at 18 and 32 weeks separately and taking the mean of depression scores at 18 and 32 weeks to explore associations with offspring cardiometabolic health across the entire distribution (to examine associations at preclinical and clinical levels).

### Study outcomes
#### Anthropometry
Body mass index (BMI: weight (kg) divided by height squared $(m^2)$) was calculated from 1 to 18 years using data from research clinics, routine offspring health clinics, health visitor records and parent-reported questionnaires. Whole body less head, and central fat and lean mass were derived from whole body dual-energy X-ray absorptiometry scans assessed five times at ages 9, 11, 13, 15 and 18 using a Lunar prodigy narrow fan beam densitometer.

#### Systolic blood pressure, diastolic blood pressure and pulse rate
At each clinic (ages 7, 9, 10, 11, 13, 15 and 18), offspring systolic blood pressure (SBP), diastolic blood pressure (DBP) and pulse rate were measured at least two times. Measures were taken when the offspring was sitting and at

rest with the arm supported, using a validated device and a cuff size appropriate for upper arm circumference. The mean of the two final measures was used here.

### Blood-based biomarkers

Insulin was measured from cord blood at birth and from research clinic samples at age 9, 15 and 18 years. Non-fasting glucose was available at age 7; fasting glucose was available at age 9, 15 and 18 years.

Triglycerides, high-density lipoprotein cholesterol (HDL-c) and total cholesterol were measured in cord blood at birth and from venous blood subsequently. Samples were non-fasted at 7 and 9 years; fasting measures were available from clinics at 15 and 18 years. Non-HDL-c was calculated by subtracting HDL-c from total cholesterol at each measurement occasion. Trajectories of blood-based biomarkers are, thus, a combination of measures from cord blood, fasting bloods and non-fasting bloods, with most measures obtained through standard clinical chemistry assays.

Further details on measurement sources for outcomes are available in online supplemental methods S2.

### Covariates

We considered the following as potential confounders: household social class, parity and maternal age, education, smoking during pregnancy, pre-pregnancy BMI, all measured by mother-completed or mother's partner-completed questionnaires; details in online supplemental methods S3. Offspring birth weight is a potential mediator of associations of prenatal anxiety and depression with later child cardiometabolic outcomes[3 6] and was not adjusted for in our analyses.

### Statistical analysis

Sex-specific patterns of change in each risk factor have been modelled previously using multilevel models[24 25] and were used here as outcome trajectories. Multilevel models estimate mean trajectories of the outcome while accounting for the non-independence (ie, clustering) of repeated measurements within individuals, change in scale and variance of measures over time and differences in the number and timing of measurements between individuals. Models include all available data from all eligible participants under a Missing at Random assumption.

Trajectories of BMI were modelled using fractional polynomials[26]; all other risk factors were estimated using linear splines (two levels for all models: measurement occasion and individual). Fractional polynomial terms and linear spline periods were selected based on model fit statistics and examination of observed data. Lean mass and fat mass included three periods of change; from 9 to 13, 13 to 15 and 15 to 18; SBP, DBP and pulse rate included three periods of change from 7 to 12, 12 to 16, and 16 to 18; HDL-c included two periods of change from 0 to 7 and from 7 to 18; non-HDL-c and triglycerides included two periods of change from 0 to 9, 9 to 18; insulin included three periods of change, from 0 to 9, 9 to 15, 15 to 18;

glucose included two periods of change from 7 to 15 and 15 to 18. Further information on the modelling of these trajectories is described in online supplemental methods S4 and tables S1–S7.

### Association between maternal prenatal anxiety and depression and trajectories of offspring cardiometabolic health outcomes

Associations between maternal prenatal anxiety and depression, and trajectories of cardiometabolic risk factors were examined by including an interaction between the categories of each exposure (primary analyses) or mean scores for continuous analyses (secondary analyses) and fractional polynomial age terms or linear spline periods. Based on previous modelling of outcomes, the mean outcome trajectory for each risk factor was allowed to vary by sex. We also explored whether associations between each prenatal maternal anxiety and depression exposure differed between women and men by including an interaction term between each exposure and sex. These analyses demonstrated no strong evidence of a sex interaction in the association of prenatal maternal anxiety and depression and cardiometabolic health outcome trajectories; thus, all analyses were subsequently performed and presented sex combined. We performed unadjusted and confounder adjusted analyses for all models.

Values of cardiometabolic risk factors that had a skewed distribution (BMI, fat mass, insulin and triglyceride) were (natural) log transformed prior to analysis; differences between each exposure category and the reference group from these models are calculated on the log scale. These values were then back transformed and are interpreted as the ratio of geometric means. Fat mass and lean mass were adjusted for height using the time and sex-varying power of height that best resulted in a height-invariant measure.[20] All trajectories were modelled in MLwiN V.3.04, using the runmlwin command in Stata V.16.

### Participants and measures included in analyses

Participants with measures of maternal prenatal anxiety and depression and at least one measure of a risk factor and complete data on all confounders were included in analyses. Offspring who reported being pregnant at the 18-year clinic (n=6) were excluded from analyses at that time point only. Online supplemental figure 1 shows a flow diagram for the study.

### Patient and public involvement

This was a secondary analysis of data from the ALSPAC birth cohort and did not involve patient and public involvement.

## RESULTS

Online supplemental table 8 shows the number of offspring with available measures of cardiometabolic risk factors at each age. The number of mother–offspring pairs included in analyses ranged from 526 participants (1464 repeated measures) for analyses of insulin to 8606

**Table 1** Characteristics of ALSPAC participants included in analysis, by maternal anxiety levels during pregnancy

| | No anxiety during pregnancy n=6137 | Anxiety at 18 weeks gestation only n=651 | Anxiety at 32 weeks gestation only n=792 | Anxiety at 18 and 32 weeks gestation n=1026 |
|---|---|---|---|---|
| | n (%) | n (%) | n (%) | n (%) |
| **Household social class*** | | | | |
| Professional | 936 (15.3) | 74 (11.4) | 87 (11.0) | 93 (9.1) |
| Managerial and technical | 2673 (43.6) | 282 (43.3) | 310 (39.1) | 408 (39.8) |
| Non-manual | 1574 (25.6) | 174 (26.7) | 222 (28.0) | 284 (27.7) |
| Manual | 677 (11.0) | 86 (13.2) | 107 (13.5) | 167 (16.3) |
| Part skilled and unskilled | 277 (4.5) | 35 (5.4) | 66 (8.3) | 74 (7.2) |
| **Maternal education** | | | | |
| Less than O level | 1410 (23.0) | 152 (23.3) | 249 (31.4) | 313 (30.5) |
| O level† | 2228 (36.3) | 257 (39.5) | 273 (34.5) | 409 (39.9) |
| A level | 1549 (25.2) | 167 (25.7) | 170 (21.5) | 207 (20.2) |
| Degree or above | 950 (15.5) | 75 (11.5) | 100 (12.6) | 97 (9.5) |
| **Mother's partner's highest educational qualification** | | | | |
| Less than O level | 1675 (28.0) | 189 (30.0) | 272 (35.9) | 352 (36.1) |
| O level† | 1320 (22.1) | 151 (24.0) | 176 (23.2) | 225 (23.1) |
| A level | 1711 (28.6) | 167 (26.6) | 196 (25.9) | 263 (26.9) |
| Degree or above | 1271 (21.3) | 122 (19.4) | 114 (15.0) | 136 (13.9) |
| **Maternal smoking during pregnancy** | | | | |
| No | 5040 (82.1) | 480 (73.7) | 587 (74.1) | 674 (65.7) |
| Yes | 1097 (17.9) | 171 (26.3) | 205 (25.9) | 352 (34.3) |
| **Parity** | | | | |
| 0 | 2795 (45.5) | 319 (49.0) | 352 (44.4) | 427 (41.6) |
| 1 | 2286 (37.2) | 220 (33.8) | 289 (36.5) | 364 (35.5) |
| 2 | 1056 (17.2) | 112 (17.2) | 151 (19.1) | 235 (22.9) |
| | **Mean (SD)** | **Mean (SD)** | **Mean (SD)** | **Mean (SD)** |
| Gestational age (weeks) | 39.5 (1.8) | 39 (1.7) | 39 (1.6) | 39 (1.8) |
| Birth weight (g) | 3431(535) | 3395(527) | 3417(500) | 3423(546) |
| Pre-pregnancy BMI (kg/m$^2$) | 22.9 (3.7) | 23 (4.2) | 23 (3.8) | 23 (4.0) |
| Maternal age (years) | 29.0 (4.5) | 28 (4.9) | 28 (5.0) | 28 (4.9) |

Number of participants available for analyses of BMI (n=8606) used as the denominator in this table given the varying sample sizes included in analyses.
*Household social class was measured as the highest of the mother's or her partner's occupational social class using data on job title and details of occupation collected about the mother and her partner from the mother's questionnaire at 32 weeks gestation. Social class was derived using the standard occupational classification codes developed by the United Kingdom Office of Population Census and Surveys and classified as I professional, II managerial and technical, IIINM non-manual, IIIM manual, and IV and V part skilled occupations and unskilled occupations
†O levels equate to current General Certificate of Secondary Education (GCSEs) demonstrating education level at aged 16.
ALSPAC, Avon Longitudinal Study of Parents and Children; BMI, body mass index.

(80 796 repeated measures) for analyses of BMI. Prevalence of anxiety was 7.5% (n=651) at 18 weeks only, 9% (n=792) at 32 weeks only and 12% (n=1026) at both 18 and 32 weeks. Prevalence of probable clinical depression was 5.5% (n=474) at 18 weeks only; 7.1% (n=610) at 32 weeks only and 6.3% (n=542) at both 18 and 32 weeks. Table 1 shows characteristics of participants by maternal prenatal anxiety levels (note, patterns of participant characteristics were similar for categories of prenatal maternal depression and are not shown here). Women with anxiety at any point during pregnancy (n=2469, 29%) were more likely to have lower education and to smoke

during pregnancy. Mother–offspring pairs included in analyses of BMI (n=8606 and maximum available sample size of all analyses) tended to have higher education and lower levels of smoking during pregnancy compared with those excluded due to missing exposure, outcome or confounder data (n=5261 to 13 341) (online supplemental table 9).

**Anxiety during pregnancy**
In confounder-adjusted analyses, associations were not observed between anxiety at 18 and 32 weeks and BMI and fat mass (figure 1, online supplemental tables 10 and

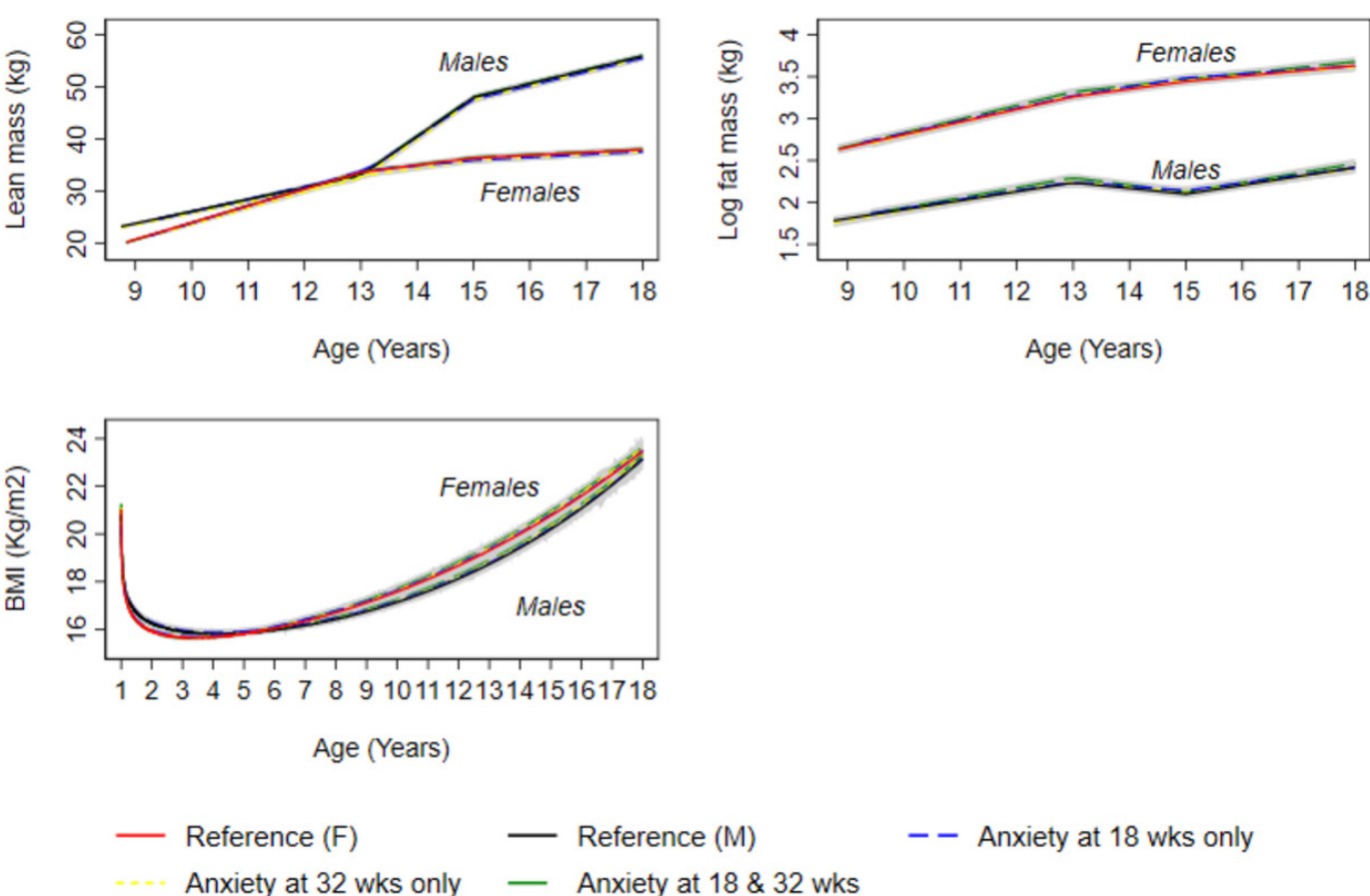

**Figure 1** Mean predicted confounder adjusted trajectories of lean mass (9–18 years), log fatmass (9–18 years) and log BMI (1–18 years), by maternal anxiety levels during pregnancy. Trajectories are adjusted for maternal age at birth of offspring, parental household socioeconomic position, parity, maternal pre-pregnancy body mass index and maternal smoking during pregnancy. Anxiety at 18 and 32 weeks gestation is defined as ≥85th percentile of the anxiety subscale of the Crown-Crisp Index for the whole cohort at each time point separately (≥85th percentile=8 at both 18 and 32 weeks gestation).

11). For instance, the difference in fat mass at 18 years between offspring of women who experienced anxiety at 18 and 32 weeks and those who did not was 4.77% (95% CI −0.51 to 10.05). Anxiety at 18 weeks only was also associated with higher DBP at 7 years in unadjusted analyses (0.70 mm Hg (95% CI 0.02 to 1.38); this difference persisted at age 18 years (difference at 18 years; 0.89 mm Hg (95% CI 0.05 to 1.73) (figure 2 and online supplemental table 12).

In confounder-adjusted analyses, we found no strong evidence that prenatal anxiety was associated with trajectories of lean mass from 9 to 18 years (figure 1 and online supplemental table 13); glucose from 7 to 18 years (figure 3 and online supplemental table 14) and insulin (figure 3 and online supplemental table S15), triglyceride (figure 4 and online supplemental material 15), HDL-c and non-HDL-c (figure 4 and online supplemental table 16), all from birth to 18 years.

### Continuous analyses

Results for all analyses were similar when prenatal maternal anxiety was examined as a continuous exposure at each time point separately or when the mean of the two measures was examined (see online supplemental figures 2-12).

### Depression during pregnancy

Associations were not observed between depression at both 18 and 32 weeks of gestation, BMI and fat mass (online supplemental figure 13, online supplemental tables 16 and 17). For instance, the difference in fat mass at 18 years for offspring of women who experienced depression at 18 and 32 weeks was 2.32% (95% CI −4.58 to 9.22). Depression at 18 weeks only was associated with higher SBP at age 18 (1.62 mm Hg (95% CI 0.17 to 3.07)) (online supplemental figure 14 and table 19). We found no strong evidence that depression during pregnancy was associated with trajectories of DBP and pulse rate from 7 to 18 (online supplemental figure 14 and table 19), lean mass from 9 to 18 years (online supplemental figure 13 and table 20), glucose from 7 to 18 years (online supplemental figure 15 and online supplemental table 21), insulin (online supplemental figure 15 and online supplemental table 22), triglyceride (online supplemental figure 16 and online supplemental table 22), HDL-c and non-HDL-c (online supplemental figure

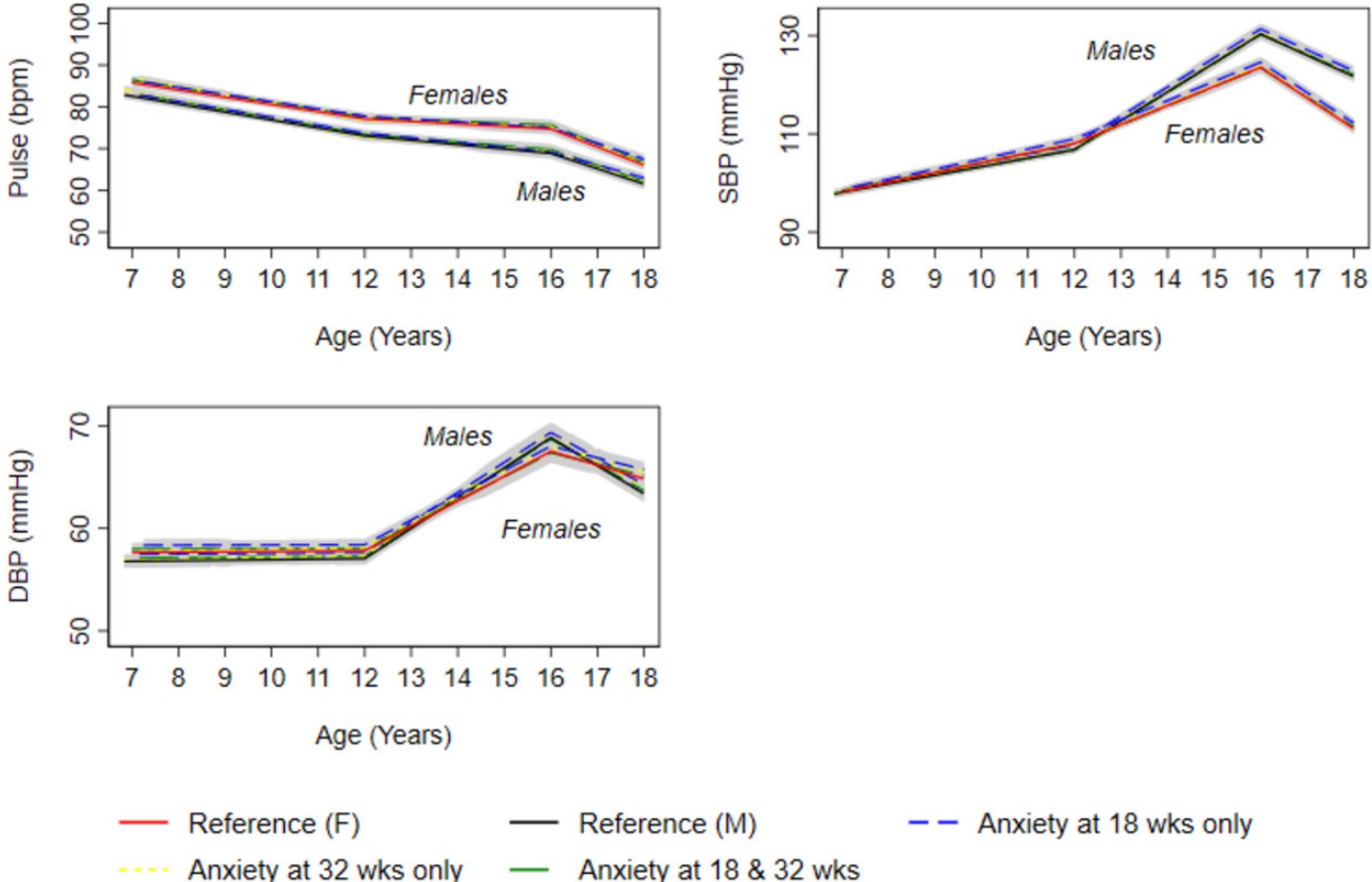

**Figure 2** Mean predicted confounder adjusted trajectories of pulse rate, SBP and DBP from 7 to 18 years, by maternal anxiety levels during pregancy. Trajectories are adjusted for maternal age at birth of offspring, parental household socioeconomic position, parity, maternal pre-pregnancy body mass index and maternal smoking during pregnancy. Anxiety at 18 and 32 weeks gestation is defined as ≥85th percentile of the anxiety subscale of the Crown-Crisp Index for the whole cohort at each time point separately (≥85th percentile = 8 at both 18 and 32 weeks gestation). DBP, diastolic blood pressure; SBP, systolic blood pressure.

16 and online supplemental table 23), all from birth to 18 years.

### Continuous analyses

Results for all analyses were similar when prenatal maternal depression was examined as a continuous exposure at each time point separately or when the mean of the two measures was examined (see online supplemental figures 17-27).

### DISCUSSION

In this large, contemporary prospective birth cohort study with repeated assessment of exposures and outcomes, we found no strong evidence that maternal anxiety and depression during pregnancy were associated with offspring cardiometabolic risk factors from birth to 18 years, regardless of whether exposure occurred at 18 weeks or 32 weeks gestation or both.

Confounder-adjusted analyses did not indicate associations between prenatal maternal anxiety and depression and offspring anthropometric outcomes. Our findings differ from previous research, suggesting maternal prenatal

anxiety and depression adversely impact offspring health and developmental outcomes.[12 13 27] A recent examination of the impact of mid-pregnancy maternal prenatal depression and anxiety identified associations with higher triglycerides in females and higher pulse rate in males at 10 years.[14] However, similar to our study, this study did not find associations between maternal prenatal depression and anxiety and offspring blood pressure, glucose, insulin or serum lipids.[14] Similarly, a number of prospective cohort studies have reported a lack of evidence to support associations between anxiety and depression and blood pressure,[15] adiposity[17–19] and glucose and insulin resistance[12]; such studies examined cardiometabolic risk factors at static time points only, however.

Attenuation of associations in adjusted analyses highlights that factors such as household social class, smoking during pregnancy and BMI influence child cardiometabolic trajectories over time. This in line with consistent evidence linking sociodemographic factors, and maternal behaviours and weight status to child cardiometabolic risk.[28–30] As such, the impact of these factors on child health outcomes likely exceeds any impact of prenatal

**Figure 3** Mean predicted confounder adjusted trajectories of HDL-c, log triglyceride and non-HDL-c from birth to 18 years, by maternal anxiety levels during pregnancy. Trajectories are adjusted for maternal age at birth of offspring, parental household socioeconomic position, parity, maternal pre-pregnancy body mass index and maternal smoking during pregnancy. Anxiety at 18 and 32 weeks gestation is defined as ≥85th percentile of the anxiety subscale of the Crown-Crisp Index for the whole cohort at each time point separately (≥85th percentile = 8 at both 18 and 32 weeks gestation). HDL-c, high-density lipoprotein cholesterol.

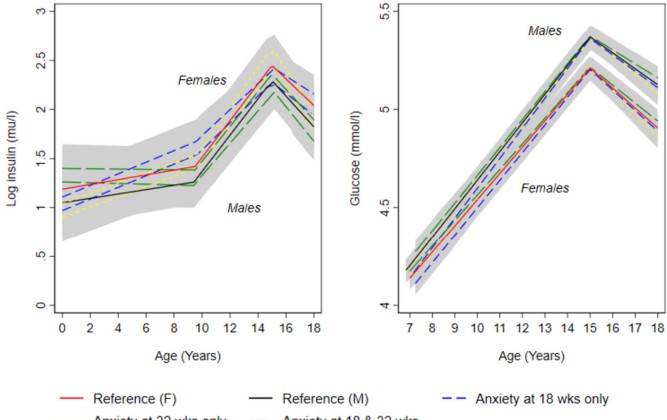

**Figure 4** Mean predicted confounder adjusted trajectories of log insulin (birth to 18 years) and glucose (7–18 years), by maternal anxiety levels during pregnancy. Trajectories are adjusted for maternal age at birth of offspring, parental household socioeconomic position, parity, maternal pre-pregnancy body mass index and maternal smoking during pregnancy. Anxiety at 18 and 32 weeks gestation is defined as ≥85th percentile of the anxiety subscale of the Crown-Crisp Index for the whole cohort at each time point separately (≥85th percentile=8 at both 18 and 32 weeks gestation).

anxiety and depression in this cohort. While such factors attenuated some observed associations and indicate potential areas for future research, we do not believe that this attenuation suggests undetected mediators of foetal programming in this study. This is because factors, such as maternal BMI in pregnancy, do not represent mediators, which sit along the causal pathway between exposure and outcome but are instead confounders that represent common causes of both maternal mental health and offspring cardiovascular risk and that typically arise prior to both exposure and outcome. Lack of observed associations in the current study may also arise because in utero maternal anxiety and depression exposure may be too early to impact on offspring cardiometabolic trajectories over the long term.[13] A recent review of associations between maternal stress and child weight outcomes found that later childhood exposure was more strongly associated with offspring weight than exposure in infancy.[31] Thus, we may not have observed associations here because maternal prenatal anxiety and depression confer little impact on offspring cardiometabolic health across the early life course, irrespective of gestational timing of

exposure. However, given our identification of some associations with offspring fat mass in this study, future work is needed to further examine effects of maternal prenatal anxiety and depression on offspring cardiometabolic health in the first two decades of life. The impact of post-natal anxiety and depression on child cardiometabolic trajectories in the ALSPAC cohort are currently being examined by the authors to further examine the effect of later exposure. Future research examining the impacts of on-going maternal mood disturbance across the perinatal period also warrants further examination.

### Strengths and limitations
Strengths of this study include prospective measurement of anxiety and depression two times during pregnancy; analysis of repeated measures of 11 key cardiometabolic risk factors from birth to 18 years; use of multilevel models accounting for clustering of repeated measures within individuals and correlation between measures over time. Examination of anxiety and depression as both categorical and continuous exposures is a further strength, which enabled examination of exposures in terms of clinical cut-offs, and incremental increases in exposure levels across the entire distribution of anxiety and depression; this is important because exposure to subclinical levels may still result in adverse outcomes that would be missed if data were examined only categorically. Limitations include generalisability of findings, as included participants tend to be more advantaged than those excluded due to missing exposure, confounder or outcome data. The range in participant numbers between analyses due to participant follow-up may represent selection bias and further impact generalisability of findings. In addition, ALSPAC includes a high proportion (~98%) white women, limiting representativeness of the findings for non-white ethnicities and ability to perform subgroup analyses for different ethnicity groups. Similarly, ALSPAC included live births only; women experiencing acute and/or chronic distress may have experienced spontaneous abortion, leading to live birth bias.[32] Self-reporting of prenatal anxiety and depression is a further limitation because self-reports do not tend to correlate well with psychophysiological indicators that could programme risk.[33] A further measurement limitation was the use of a generalised anxiety measure in the cohort rather than inclusion of pregnancy-specific anxiety measure. Pregnancy-specific anxiety is distinct from general anxiety[34] and is a robust risk factor for child health outcomes, beyond the impact of general anxiety and depression.[29 35] As such, future research should include a pregnancy-specific measure to determine potential differential effects on child cardiometabolic outcomes. In addition, the role of additional factors, such as prenatal antidepressant use, could not be examined in the current study due to the low proportion of women reporting antidepressant use in the cohort; future research should consider the role of factors such as antidepressant use in

potential associations between maternal mental health and child cardiometabolic health outcomes.

## CONCLUSION
Our findings suggest that maternal prenatal anxiety and depression do not impact offspring cardiometabolic health outcomes in the first two decades of life; these findings may provide reassurance to women experiencing prenatal anxiety and/or depression that any impacts on offspring cardiometabolic health from birth to the end of adolescence are likely to be small. Approaches and strategies to prevent and/or reduce maternal prenatal anxiety and depression may have limited impact on offspring cardiometabolic health in the first two decades of life.

**Author affiliations**
[1]School of Public Health, University College Cork, Cork, Ireland
[2]Population Health Sciences, Bristol Medical School, University of Bristol, Bristol, UK
[3]School of Medicine, Trinity College Dublin, Dublin, Ireland
[4]Department of Clinical, Neuro- and Developmental Psychology, VU University Amsterdam, Amsterdam, Netherlands
[5]Irish Centre for Maternal and Child Health Research (INFANT) Centre, Cork University Maternity Hospital, Cork, Ireland
[6]School of Health Sciences, University of Nottingham, Nottingham, UK
[7]MRC Integrative Epidemiology Unit, University of Bristol, Bristol, UK

**Acknowledgements** We are extremely grateful to all the families who took part in this study, the midwives for their help in recruiting them, and the whole ALSPAC team, which includes interviewers, computer and laboratory technicians, clerical workers, research scientists, volunteers, managers, receptionists and nurses. We are also grateful to Dr Darren Dahly in the Health Research Board Clinical Research Facility Cork for feedback on an earlier draft of this manuscript.

**Contributors** KM-S conceptualised the study, conducted analyses, drafted and revised the manuscript. KON conducted analyses, and critically reviewed and revised the manuscript. AF critically reviewed and revised the manuscript. CH contributed to interpretation of findings and critically reviewed and revised the manuscript. LH critically reviewed and revised the manuscript. ACH contributed to interpretation of findings and critically reviewed and revised the manuscript. PMK contributed to interpretation of findings and critically reviewed and revised the manuscript. AK contributed to interpretation of findings and critically reviewed and revised the manuscript. SAR contributed to interpretation of findings and critically reviewed and revised the manuscript. LMO'K conceptualised the study, conducted analyses, drafted and revised the manuscript. All authors approved the final manuscript. KMS and LOK are responsible for the overall content as the guarantors.

**Funding** The UK Medical Research Council and Wellcome (Grant ref: 217065/Z/19/Z) and the University of Bristol provide core support for ALSPAC. This publication is the work of the authors and KMS and LMOK will serve as guarantors for the contents of this paper. KMS is supported by a Health Research Board of Ireland (HRB) Applying Research into Policy and Practice award (HRB-ARPP- A011). LMOK and KON are supported by a HRB Emerging Investigator Award (EIA-2019-007). KON is also supported by a HRB Research Leader Award (HRB RL/2013/7) and a HRB Interdisciplinary Capacity Enhancement Award (HRB ICE 2015-1026). LDH and AF are supported by Career Development Awards from the UK Medical Research Council (grants MR/M020894/1 and MR/M009351/1, respectively). LDH and AF work in a unit that receives funds from the UK Medical Research Council (grant MC_UU_00011/3, MC_UU_00011/6). The research funders had no role in the study design; data collection, analysis, and interpretation of data; writing of the manuscript; or the decision to submit the manuscript for publication.

**Competing interests** None declared.

**Patient consent for publication** Not applicable.

**Ethics approval** This study used secondary data from the ALSPAC cohort. Participants in the original cohort study provided informed consent to participate and for their data to be used as secondary data in subsequent research.

**Provenance and peer review** Not commissioned; externally peer reviewed.

**Data availability statement** Data may be obtained from a third party and are not publicly available. Data used in this study was from the Avon Longitudinal Study of Parents and Children (ALSPAC), requests for access to this data can be made to the ALSPAC executive committee.

**ORCID iDs**
Karen Matvienko-Sikar http://orcid.org/0000-0003-2777-6581
Kate O' Neill http://orcid.org/0000-0003-4843-4265
Catherine Hayes http://orcid.org/0000-0002-1576-4623
Patricia M Kearney http://orcid.org/0000-0001-9599-3540

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
