## [Reviewer comments · BMJ Open]

ARTICLE DETAILS

TITLE (PROVISIONAL)	Maternal prenatal anxiety and depression and trajectories of cardiometabolic risk factors across childhood and adolescence: a prospective cohort study
AUTHORS	Matvienko-Sikar, Karen; O'Neill, Kate; Fraser, Abigail; Hayes, Catherine; Howe, Laura; Huizink, A. C.; Kearney, Patricia; Khashan, Ali; Redsell, Sarah A.; O'Keeffe, Linda M

VERSION 1 – REVIEW

REVIEWER	Oberlander, Tim University of British Columbia School of Population and Public Health
REVIEW RETURNED	02-May-2021

GENERAL COMMENTS	Using a large prospective birth cohort this paper reports on associations between maternal prenatal anxiety and depression and trajectories of offspring cardiometabolic risk during the first two decades of life. Unadjusted analyses suggested associations between maternal mood and anthropometric outcomes (ie BMI, fat mass), these associations attenuated when accounting for confounders (maternal age at delivery, smoking maternal BMI etc). While these findings are in contrast to previous studies, this study offers a number of key strengths. These include its large birth cohort prospective design, use of multiple measures that enabled the use of repeated measure multilevel models and use of both dichotomous and continuous measures of prenatal maternal mood as prenatal exposures. Granted accounting for negative findings are challenging, however the paper could be strengthened by a more robust discussion of what might account for why the confounders attenuated what appeared to be a 'main effect' finding. Namely, could this finding represent yet undetected mediators that underlie possible 'fetal programming' associations between maternal mood and cardiovascular risks in offspring? If it's not possible to undertake mediator analyses, could the authors propose possible further studies to further examine what might be arguably their most interesting finding (ie the effect of confounders)? Further, was there an attempt to examine mediator models (ie was the effect of prenatal maternal mood on child outcomes mediated by child physical activity - or other known risk factors for BMI). What role did on going maternal mood disturbances (postnatal) play in childhood CVS risk? (i.e. prenatal mood disturbances predict post natal mood disturbances).
---

	Was there an effect of prenatal maternal treatment with antidepressants play? What happens when maternal anxiety and depression measure were considered as a composite measure (ie internalizing symptoms) rather than a discreet mood dimension?
--	---

REVIEWER	Coussons-Read, Mary University of Colorado
REVIEW RETURNED	04-Oct-2021

GENERAL COMMENTS	This is a straightforward, logical, and important analysis of the potential impact of maternal prenatal anxiety and depression on cardiometabolic risk factors in offspring up to 18 years of age. The authors have leveraged a large, well-collected dataset to examine these questions although some limitations must be addressed prior to publication.  1. The number of live births exceeds the number of women who provided informed consent. This suggests that the study was not limited to Singleton pregnancies, but detail about this is not provided. Please clarify. 2. A concern is the anxiety measure assessed in the present study. A significant body of research indicates that generalized anxiety does not have a significant impact on pregnancy and birth outcome relative to pregnancy-specific anxiety. Although these findings were not available at the time of prenatal data collection in the study, addressing this as a potential limitation in the discussion would be helpful. 3. Given the appropriateness of the confounders considered in the statistical analysis, there is no reason to report statistics that do not account for these confounders. Discussion of the statistical results should be limited to analyses that control for the identified confounding variables. 4. There are some grammatical and syntactic errors throughout the paper which can be addressed with careful proofreading.
--

VERSION 1 – AUTHOR RESPONSE

Reviewer: 1

Dr. Tim Oberlander, University of British Columbia School of Population and Public Health

Comments to the Author:

Reviewer Comment 1.

Using a large prospective birth cohort this paper reports on associations between maternal prenatal anxiety and depression and trajectories of offspring cardiometabolic risk during the first two decades of life. Unadjusted analyses suggested associations between maternal mood and anthropometric outcomes (ie BMI, fat mass), these associations attenuated when accounting for confounders (maternal age at delivery, smoking maternal BMI etc). While these findings are in contrast to previous studies, this study offers a number of key strengths. These include its large birth cohort prospective design, use of multiple measures that enabled the use of repeated measure multilevel models and use of both dichotomous and continuous measures of prenatal maternal mood as prenatal exposures.

Granted accounting for negative findings are challenging, however the paper could be strengthened by a more robust discussion of what might account for why the confounders attenuated what appeared to be a 'main effect' finding. Namely, could this finding represent yet undetected mediators that underlie possible 'fetal programming' associations between maternal mood and cardiovascular risks in offspring? If it's not possible to undertake mediator analyses, could the authors propose possible further studies to further examine what might be arguably their most interesting finding (ie the effect of confounders)?

Further, was there an attempt to examine mediator models (ie was the effect of prenatal maternal mood on child outcomes mediated by child physical activity - or other known risk factors for BMI).

Author Response 1.

Thank you for your comments and suggestions to improve the manuscript.

We agree that the role of the confounders in attenuating the findings is interesting and important and have included further discussion of the role of the confounders in the discussion section on page 16 as outlined below.

Attenuation of associations in adjusted analyses highlights that factors such as household social class, smoking during pregnancy, and BMI influence child cardiometabolic trajectories over time. This is in line with consistent evidence linking sociodemographic factors, and maternal behaviours and weight status to child cardiometabolic risk [28–30]. As such, the impact of these factors on child health outcomes likely exceeds any impact of prenatal anxiety and depression in this cohort.

We have also added the following additional text to the discussion on page 17 to clarify the role of confounders, as we do not believe we can infer that the attenuation by the confounders suggests undetected mediators of foetal programming in our study.

While such factors attenuated some observed associations and indicate potential areas for future research, we do not believe this attenuation suggests undetected mediators of foetal programming in this study. This is because factors, such as maternal BMI at the end of pregnancy, do not represent mediators which sit along the causal pathway between exposure and outcome, but are instead confounders that represent common causes of both maternal mental health and offspring cardiovascular risk and that typically arise prior to both exposure and outcome.

Given the lack of evidence of association of exposure and outcome after adjustment for confounders, we have not conducted further mediator analyses, because there is no total effect of prenatal depression and anxiety and outcome to decompose into direct and indirect effects using mediation analyses.

Reviewer comment 2.

What role do ongoing maternal mood disturbances (postnatal) play in childhood CVS risk? (i.e. prenatal mood disturbances predict post natal mood disturbances).

Author Response 2.

This is an important point and one that we are examining in a separate paper, where we have identified some impacts of postnatal anxiety and depression on cardiometabolic health outcomes. We chose to examine the prenatal and postnatal periods separately because, while there is some evidence of associations between the two periods for maternal mental health, the potential mechanisms by which cardiometabolic outcomes are likely to occur are expected to be different (e.g. foetal programming as a mechanism for prenatal mood versus parenting behaviours for postnatal mood.) We have added a note on this to the discussion section on page 16 as follows:

The impact of postnatal anxiety and depression on child cardiometabolic trajectories in the ALSPAC cohort is currently being examined by the authors to further examine the effect of later exposure. Future research examining the impacts of on-going maternal mood disturbance across the perinatal period also warrants further examination.

Reviewer comment 3

Was there an effect of prenatal maternal treatment with antidepressants play?

Author response 3

The role of antidepressants in prenatal mood is important, however we chose not to include that variable in our analyses due to the low numbers of women within the cohort who reported antidepressant use at either/both 18 and 32 weeks gestation. As identified by Headley et al. (2004), in a sample of 13,194 pregnancies at 18 weeks, psychoanaleptics were used in 0.002% of pregnancies and psycholeptics were used in 0.006% of pregnancies. At 32 weeks, psychoanaleptics were used in 0.001% of pregnancies, while psycholeptics were used in 0.007% of pregnancies. Given this lower number of participants, we would have lacked sufficient power to examine any potential effects in this study.

Given the importance of antidepressants during pregnancy, we have added the following to the limitations section in the discussion, on pages 17 and 18:

In addition, the role of additional factors, such as prenatal antidepressant use could not be examined in the current study due to the low proportion of women reporting antidepressant use in the cohort; future research should consider the role of factors such as antidepressant use in potential associations between maternal mental health and child cardiometabolic health outcomes.

Reviewer comment 4

What happens when maternal anxiety and depression measure were considered as a composite measure (ie internalizing symptoms) rather than a discreet mood dimension?

Author response 4

Anxiety and depression were measured separately, rather than as a composite measure because, although anxiety and depression are highly related constructs, they are conceptually different. For instance, anxiety and depression differ in terms of cognitive processes and psychophysiological responses, as outlined in more detail by Eysenck and Fajkowska (2018). As such, we chose not consider anxiety and depression as a composite measure as this would impede examination of potential differential impacts of these constructs on child health outcomes.

Reviewer: 2

Dr. Mary Coussons-Read, University of Colorado

Comments to the Author:

This is a straightforward, logical, and important analysis of the potential impact of maternal prenatal anxiety and depression on cardiometabolic risk factors in offspring up to 18 years of age. The authors have leveraged a large, well-collected dataset to examine these questions although some limitations must be addressed prior to publication.

Author response

Thank you for your comments and feedback, which we have now incorporated into the paper.

Reviewer comment 1.

1. The number of live births exceeds the number of women who provided informed consent. This suggests that the study was not limited to Singleton pregnancies, but detail about this is not provided. Please clarify.

Author Response 1.

Apologies for the lack of clarity on this point. We have now updated Figure 1 in the supplemental file to better reflect that 14,451 pregnant women were recruited, resulting in 13,867 pregnancies with live births. Of these, 106 women had a second pregnancy and were excluded and so 13,761 women with pregnancies that resulted in live births were included. Approximately 2.8% of these pregnancies were multiple pregnancies and all pregnancies were included in our analyses. This inclusion has helped to clarify the previous confusion within this figure, thank you.

Reviewer comment 2.

2. A concern is the anxiety measure assessed in the present study. A significant body of research indicates that generalized anxiety does not have a significant impact on pregnancy and birth outcome relative to pregnancy-specific anxiety. Although these findings were not available at the time of prenatal data collection in the study, addressing this as a potential limitation in the discussion would be helpful.

Author response 2.

We agree that the lack of a pregnancy-specific measure is a limitation of the study and this has been included in the discussion on page 17 as follows:

A further measurement limitation was the use of a generalized anxiety measure in the cohort rather than inclusion of pregnancy-specific anxiety measure. Pregnancy-specific anxiety is distinct from general anxiety[31] and is a robust risk factor for child health outcomes, beyond the impact of general anxiety and depression[32,33]. As such, future research should include a pregnancy-specific measure to determine potential differential effects on child cardiometabolic outcomes.

Reviewer comment 3.

3. Given the appropriateness of the confounders considered in the statistical analysis, there is no reason to report statistics that do not account for these confounders. Discussion of the statistical results should be limited to analyses that control for the identified confounding variables.

Author response 3.

Discussion of unadjusted analyses has been removed from the results section, which now presents the adjusted statistical results, while unadjusted findings are retained in the respective tables for reference.

This change is also reflected in the results section of the abstract, as follows:

Results. We found no strong evidence of associations between maternal anxiety and depression, and offspring trajectories of cardiometabolic risk factors. Depression at 18 weeks was associated with higher SBP at age 18 (1.62 mmHg (95% CI, 0.17, 3.07)). Anxiety at 18 weeks was also associated with higher DBP at 7 years in unadjusted analyses (0.70 mmHg (95% CI, 0.02, 1.38)); this difference persisted at age 18 years (difference at 18 years; 0.89 mmHg (95% CI, 0.05, 1.73)). No associations were observed for body mass index; fat mass; lean mass; pulse rate; glucose; triglycerides, high-density lipoprotein cholesterol and non-high-density lipoprotein cholesterol, and insulin.

In addition, we removed reference to the unadjusted analyses in the discussion on page 15 as follows:

Confounder adjusted analyses did not indicate associations between prenatal maternal anxiety and depression and offspring anthropometric outcomes.